# Prone Positioning in Mechanically Ventilated COVID-19 Patients: Timing of Initiation and Outcomes

**DOI:** 10.3390/jcm12134226

**Published:** 2023-06-23

**Authors:** Alexander Jackson, Florence Neyroud, Josephine Barnsley, Elsie Hunter, Ryan Beecham, Meiarasu Radharetnas, Michael P. W. Grocott, Ahilanandan Dushianthan

**Affiliations:** 1NIHR Biomedical Research Centre, University Hospital Southampton and University of Southampton, Southampton SO16 6YD, UK; alexander.jackson@soton.ac.uk (A.J.); a.dushianthan@soton.ac.uk (A.D.); 2General Intensive Care Unit, University Hospital Southampton, Southampton SO16 6YD, UK; florence.neyroud@uhs.nhs.uk (F.N.); josephine.barnsley@gmail.com (J.B.); elsiehunter@icloud.com (E.H.); ryan.beecham@uhs.nhs.uk (R.B.); radharetnas.meiarasu@uhs.nhs.uk (M.R.)

**Keywords:** COVID-19, prone, mechanical ventilation, critical care

## Abstract

The COVID-19 pandemic led to a broad implementation of proning to enhance oxygenation in both self-ventilating and mechanically ventilated critically ill patients with acute severe hypoxic respiratory failure. However, there is little data on the impact of the timing of the initiation of prone positioning in COVID-19 patients receiving mechanical ventilation. In this study, we analyzed our proning practices in mechanically ventilated COVID-19 patients. There were 931 total proning episodes in 144 patients, with a median duration of 16 h (IQR 15–17 h) per proning cycle. 563 proning cycles were initiated within 7 days of intubation (early), 235 within 7–14 days (intermediate), and 133 after 14 days (late). The mean change in oxygenation defined as the delta PaO_2_/FiO_2_ ratio (ΔPF) after the prone episode was 16.6 ± 34.4 mmHg (*p* < 0.001). For early, intermediate, and late cycles, mean ΔPF ratios were 18.5 ± 36.7 mmHg, 13.2 ± 30.4 mmHg, and 14.8 ± 30.5 mmHg, with no significant difference in response between early, intermediate, and late proning (*p* = 0.2), respectively. Our findings indicate a favorable oxygenation response to proning episodes at all time points, even after >14 days of intubation. However, the findings cannot be translated directly into a survival advantage, and more research is needed in this area.

## 1. Introduction

The SARS-CoV-2 virus (coronavirus disease 19; COVID-19) was declared a global pandemic by the World Health Organization (WHO) in early 2020 and has infected over 750 million cases to date, resulting in over 6 million deaths [1]. The virus is implicated as a cause of diffuse alveolar damage resulting in acute hypoxic respiratory failure (AHRF) with an associated mortality of between 30 and 50% of those requiring mechanical ventilation [2,3]. The evolving nature of the pandemic has necessitated rapid development of treatment protocols, vaccination, new targeted therapies, and collaboration in global research, which has led to reduced severity of infections and improvements in mortality rates for those most severely affected [4,5]. However, mortality and long-term morbidity from the critical illness caused by COVID-19 remain significantly high, with ongoing challenges due to phenotypic variation and the emergence of new variants resulting in varying disease severity and treatment response [6,7,8,9].

For several decades, it has been recognized that positional changes improve oxygenation in mechanically ventilated patients with acute respiratory distress syndrome (ARDS) [10]. The 2015 Cochrane review of prone positioning in mechanically ventilated ARDS patients found no significant improvement in outcome except for those with severe hypoxemia when introduced early with prolonged use [11]. The PROSEVA multicenter randomized-controlled trial demonstrated a 90-day mortality reduction in mechanically ventilated severe ARDS patients who received early prone positioning for >16 h a day in comparison to the supine control group [12]. Improved gas exchange is attributed to improved lung recruitment and ventilation-perfusion matching, with the added benefit of reducing airway pressure and stress, resulting in ventilator-associated lung injury [13].

A recent systematic review and meta-analysis of awake-prone positioning in self-ventilating patients suggests a reduction in the need for endotracheal intubation, particularly in those who require advanced respiratory support [14]. The evidence for proning mechanically ventilated COVID-19 patients relies on observational studies and meta-analyses of these [15,16]. A randomized control trial comparing proned and non-prone intubated COVID-19 patients is unlikely to gain ethical approval considering existing evidence that proning is beneficial in patients with severe ARDS. More extensive observational studies are therefore crucial to widening our evidence base. However, the effect of prone positioning timing on oxygenation improvement has not been explored previously.

## 2. Materials and Methods

This was a retrospective observational single-center study conducted at University Hospital Southampton, a 1200-bed teaching hospital with 32 general intensive care unit (GICU) beds. Eligible participants were adult patients admitted to the GICU requiring mechanical ventilation and prone positioning as part of their management who were reverse transcriptase polymerase chain reaction (RT-PCR) positive for SARS-CoV-2 with the clinical syndrome of COVID-19 pneumonitis. Patients were enrolled between March 2020 and March 2021, incorporating the first two epidemiologically distinct waves of COVID-19 in the UK. Local guidance was developed for all COVID-19-related ICU admissions, and the Intensive Care Society (UK) guidelines on prone position were implemented [17]. Mechanical ventilation was instituted for all deteriorating patients, and early prone position was recommended to improve oxygenation. The study data were collected as part of the REACT COVID observational study (a prospective longitudinal cohort study reviewing clinical progression and management of COVID-19 infection from hospital admission to discharge) with ethical approval REC reference 17/NW/0632, SRB reference number SRB0025 [18]. Consent was waived due to the retrospective, observational nature of this study.

Data were collected from an electronic clinical information system (CIS) (MetaVision) using semi-automated and manual data extraction. Baseline demographic (age, gender) and clinical (symptom duration, critical illness severity score, comorbidities, and laboratory results) information was collected. Comorbidities were numerically quantified using Charlson’s comorbidity index [19]. Other ICU-specific severity indices (Sequential Organ Failure Assessment (SOFA) scores and the Acute Physiology and Chronic Health Evaluation (APACHE II)) were also collected on admission to the ICU [20,21]. All ventilatory parameters, vital signs, blood gas analysis, laboratory tests, and positional data were extracted electronically from the CIS. These were subsequently imported into R (R Core Team, Vienna, Austria), where a careful process of quality assurance and outlier management was undertaken.

The overall clinical management and the mode of mechanical ventilation were largely at the discretion of the treating physician. However, the standard mode of ventilation was adaptive pressure ventilation-controlled mechanical ventilation (APVcmv) with tidal volumes targeted at 6 mls/kg predicted body weight (PBW). The positive end expiratory pressure (PEEP) was adjusted in accordance with the ARDSnet PEEP-FiO_2_ table [22]. We used airway pressure release ventilation (APRV) as a rescue mode in patients with refractory hypoxemia [23]. Patients were sedated with infusions of fentanyl and either midazolam or propofol. We encouraged the use of neuromuscular blocker cisatracurium infusion titrated according to the train-of-four measurements during the first 48 hours. Further use of neuromuscular blockade was based on clinical need and the degree of patient-ventilatory dysynchrony. Patients with refractory hypoxemia who were suitable for extracorporeal membrane oxygenation (ECMO) were referred to the regional ECMO center according to the UK national guidelines [24].

The primary outcome is the change in arterial oxygen partial pressure (PaO_2_ in mmHg) to fractional inspired oxygen (FiO_2_) ratio (ΔP/F) with each prone cycle and changes over sequential proning cycles for everyone. We further sub-categorized the timing of the initiation of prone cycles as early (<7 days of mechanical ventilation), intermediate (7–14 days), and late (>14 days) from the initiation of mechanical ventilation.

All included patients had COVID-19 acute hypoxemic respiratory failure. There were no specific exclusion criteria applied, as long as there were adequate personnel to perform the prone position maneuvers safely. Although not mandated due to variability in the availability of resources, our local guidance advocated a prone position for patients requiring a FiO_2_ of 0.6 or above to maintain a PaO_2_ of 60 mmHg (i.e., a PaO_2_/FiO_2_ ratio of <100 mmHg). Position data were recorded hourly by the bedside nurse (including prone and supine positions) as part of routine care. From this data, the number of proning cycles, timing, and total duration of proning were obtained for individual patients. For each cycle, defined by a continuous period in either a supine or prone position, physiological parameters and arterial blood gas results were extracted. To allow comparison between adjacent prone and supine periods, an appropriate summary statistic was sought. Individual cycles were comparatively short with a limited pool of arterial blood gases to examine, and the values were non-normally distributed. Therefore, the median value of each variable (PaCO_2_, PaO_2_/FiO_2_, etc.) was used to calculate changes between adjacent cycles. Analysis of the ΔPaO_2_/FiO_2_ values demonstrated a normal distribution across the entire cohort and allowed subsequent analysis to treat these as such, hence the use of mean and standard deviations.

Quality assurance and outlier management were multi-stage processes. Initial visualization of the data allowed the identification of significant areas of error. Clinicians entered variables manually or recorded them automatically from devices, e.g., blood gas analyzers. We used cross-checking with other data sources for manually derived variables, such as FiO_2_. In the case of FiO_2_, we used ventilator FiO_2_ measurements directly downloaded from the ventilators. This allowed the correction of obvious errors, such as where the FiO_2_ had been entered as 0.21 on blood gas measurements. We also identified common errors that could occur due to the user interface (UI) of the electronic patient record (EPR). For example, both ‘PRONE’ and ‘SUPINE’ could be recorded simultaneously. Where this occurred, the values were compared to the previous and subsequent values, and the most appropriate was chosen for imputation. For automatically collected variables, device error was considered. Biologically impossible values, such as a PaO_2_ of zero, were removed. While biologically implausible values were closely examined and removed where appropriate. In cases of error, the data were removed and alternative values were not imputed.

All statistical testing was undertaken in R (R Core Team, Vienna, Austria). Continuous variables were expressed as median and interquartile range, where non-normally distributed, and as mean and standard deviation, where normally distributed. Nominal variables were summarized as numbers and percentages. For comparison of multiple groups where the data were non-normally distributed, Kruskal-Wallis’s rank sum was used. For normally distributed variables, a one-way ANOVA was used.

## 3. Results

### 3.1. Patient Demographics

There were 184 SARS-CoV-2 RT-PCR-positive mechanically ventilated patients with COVID-19 pneumonitis admitted between March 2020 and March 2021. Of those, 144 (78.3%) received one or more prone cycles. The median age was 59 years (IQR 51–66), and two-thirds were male. The median duration of symptoms prior to ICU admission was 7 days, and the median time to intubation from ICU admission was 18 h (IQR 1.6–75.2). On admission, all patients met the criteria for moderate-to-severe ARDS. The median admission APACHE II, SOFA, and Charlson Comorbidity Index were 15 (IQR 11, 23), 4 (3.0, 6.3), and 2 (1, 3), respectively. The admission PaO_2_/FiO_2_ ratio was 114.8 mmHg (IQR 87.0, 141.0), suggesting moderate to severe hypoxemia at presentation to the ICU. The median BMI was 31 kg/m^2^ (IQR 26, 36), and 55% of patients were classified as obese with a BMI of >30 kg/m^2^. The most common comorbidity was diabetes mellitus (31%), followed by chronic respiratory illnesses (20%). Demographic and baseline clinical data are presented in detail in Table 1.

### 3.2. Prone Cycle Characteristics

A total of 931 proning cycles were performed, of which 776 had sufficient data to calculate changes in blood gas variables. Most prone cycles were conducted within 7 days (early) (563, 60%), followed by 7–14 days (intermediate) (235, 25%), and >14 days (late) (133, 14%). The median duration of proning cycles was 16 h (IQR 15–17), which remained consistent across all time points (Table 2). The median number of proning cycles per patient was 5 (3–9), with one patient being proned for a total of 22 cycles. The number of patients proned and the required number of prone cycles are depicted in Figure 1. Table 2 summarizes the characteristics of proning cycles.

### 3.3. Outcome: Change in Oxygenation (∆PaO_2_/FiO_2_)

The mean change in PaO_2_/FiO_2_ ratio (∆PaO_2_/FiO_2_ or P/F) and PaCO_2_ (∆PaCO_2_) following each proning cycle was 16.6 ± 34.4mmHg and 2.2 ± 7.4 mmHg, respectively. There was no significant difference in the ∆P/F between the subgroups according to the timing of the proning cycles. The mean ∆P/F for early, intermediate, and late proning cycles were 18.5 ± 36.7 mmHg, 13.2 ± 30.4 mmHg, and 14.8 ± 30.5 mmHg, respectively. The consistent improvement in ∆P/F across all subgroups implies that proning cycles resulted in an improvement in the P/F ratio regardless of proning initiation timing. PaCO_2_ appears to increase following proning, as demonstrated by consistent positive ΔPaCO_2_; however, the value is very small and may not be clinically significant. Again, no difference was seen based on the timing of the proning cycle. Gas exchange variables are summarized in Table 3. Figure 2 demonstrates the change in P/F ratio by proning cycle number (2a) and cycle timing (2b) for each proning episode, respectively.

We also performed sub-group analysis for individual proning cycles up to 10 cycles to assess the change in oxygenation. There was a consistent improvement in oxygenation throughout the proning cycles, with the most prominent improvement in oxygenation noted for the first proning episode (Table 4).

### 3.4. Clinical Outcomes

The median duration of mechanical ventilation was 397 (IQR 242, 746) hours. Among those patients who were mechanically ventilated and required prone positioning, the 30-day hospital survival and overall hospital survival to discharge were 72% and 65%, respectively (Table 5).

## 4. Discussion

Prone positioning is common practice in mechanically ventilated patients suffering from acute hypoxaemic respiratory failure or acute respiratory distress syndrome [11,25,26]. This study looked at the effect of proning cycles on oxygenation (∆PaO_2_/FiO_2_ or P/F ratio) in mechanically ventilated COVID-19 patients after each proning episode. We sub-grouped and analyzed patients based on the timing of the initiation of proning cycles from the initiation of mechanical ventilation. These were arbitrarily classified as early (<7 days of mechanical ventilation), intermediate (7–14 days), and late (>14 days). Overall, proning cycles were associated with consistent improvements in oxygenation and small but statistically significant increases in hypercarbia.

Consistent with previously published data on COVID-19 mechanically ventilated patients, improvements in oxygenation were similar regardless of when the proning cycles were implemented (early vs. intermediate vs. late) [27]. However, a study comparing COVID-19 ARDS with historical non-COVID-19 ARDS found that delaying the first prone session for more than 24 h after intubation was associated with less oxygenation improvement [28]. In contrast, we have shown consistent improvement in oxygenation regardless of the timing of prone sessions. The variation in the analytical methods may explain the differences in results. Our study examined the details of each prone cycle response for all prone episodes rather than the oxygenation response for only the first prone session. For sustained oxygen improvement, many proning cycles may be required. The median number of proning cycles performed in our patient cohort was 5 (IQR: 3, 9). However, there was substantial variation across individuals, with one patient having 22 prone cycles. In a similar retrospective case series, compared to ARDS, patients with COVID-19-induced ARDS needed more proning sessions [29].

We studied the oxygenation data for each proning cycle up to ten cycles and found an improvement in oxygenation ranging from 5.5 to 26.8 mmHg. However, the first proning cycle showed the most improvement in oxygenation (∆P/F of 26.8 ± 48.4 mmHg). There is limited literature on the number of cycles a patient may require and the ideal time to perform a prone position, which is frequently guided by clinical response. While most patients will recover over time, some may require lengthy proning sessions to improve their oxygenation, and evidence for continuing beyond the first few days is scarce, particularly for COVID-19 patients. As far as we are aware, this is the first study to demonstrate that even after 14 days of mechanical ventilation, prone positioning may be beneficial in improving oxygenation in COVID-19 patients.

Prone positioning has been adopted as an evidence-based intervention to improve outcomes in patients with moderate to severe ARDS [30]. The improved oxygenation may be due to several mechanisms, including improved ventilation-perfusion matching, increased expiratory lung volume, and moderation of ventilator-induced lung injury by improving lung recruitability while minimizing lung overinflation [13,25]. The seminal randomized controlled trial of severe ARDS patients demonstrated that 16 h of proning improved survival, which has been supported by subsequent meta-analyses [11,12]. However, in most ARDS studies, the total duration of prone positioning was <7 days [11,12,26], and the clinical efficacy of the prone position after seven days in mechanically ventilated COVID-19 patients remains unclear. We continued to perform proning cycles until the patient was clinically improved and no longer required a FiO_2_ of 0.6 or above to maintain a PaO_2_ of 60 mmHg (i.e., a PaO_2_/FiO_2_ of >100 mmHg) and observed an on-going benefit with this approach. Patients were also reintroduced to further cycles of prone positioning when there was clinical deterioration with worsening hypoxemia, regardless of the preceding duration of mechanical ventilation.

The requirement for longer proning sessions is consistent with other published observational studies [29,31]. However, this practice differs from previously published clinical trials of ARDS patients, where most patients only had proning cycles during the early stages of their disease [26]. Another observational study have shown sustained responses to prolonged (beyond 16 h) prone positioning in mechanically ventilated COVID-19 patients [32]. A randomized control trial of 52 mechanically ventilated COVID-19 patients with prolonged proning for 24 h showed that this was safe and demonstrated improved outcomes [33]. However, this study did not directly compare the beneficial effects of 16 h with 24 h. Observational studies also concur with this RCT, where an extended duration of prone positioning beyond 16 h may confer better clinical outcomes with a reduction in staff demands but at the expense of an increased risk of prone-related complications [34,35,36]. In comparison, we opted to provide multiple 16-h proning cycles rather than longer proning cycles to minimize pressure-related complications. Larger RCTs are needed to explore the utility of prone positioning beyond 16 h.

Awake-prone positioning for COVID-19 patients with hypoxemic respiratory failure requiring advanced respiratory support (e.g., high-flow nasal oxygen therapy or non-invasive ventilation) reduces the need for invasive mechanical ventilation but not mortality [14]. However, the clinical effects of prone positioning on mechanically ventilated COVID-19 patients are primarily translated from observational studies, and randomized controlled trials are lacking. Similar to our study finding, a systematic review of observational studies concluded better oxygenation with prone ventilation than supine position [16]. In our cohort, 78% of mechanically ventilated patients required at least one prone episode, and this proportion is similar to previously published data [37,38]. The improved oxygenation from prone positioning in COVID-19 patients appears to be due to improved ventilation perfusion matching rather than significant alveolar recruitment [28,39]. Using electrical impedance tomography, Zarantonello et al. demonstrated that in COVID-19 patients, early prone position improves dorsal ventilation and reduces lung overdistention without significant changes in static lung compliance or driving pressure [39].

Prone positioning has been extensively implemented to improve oxygenation in spontaneously ventilated and mechanically ventilated COVID-19 patients. The procedure often requires multiple highly skilled personnel to deliver safely and minimize complications such as inadvertent extubation and the removal of invasive vascular access lines during the procedure. Our center developed a dedicated prone team following rapid, extensive training to optimize care delivery and minimize complications. A locally produced protocol and checklist modified from the Intensive Care Society’s UK guidelines were developed and adhered to during each proning cycle, and the interval position changed every 4 h [17].

Our study has several limitations. This is a single-center, retrospective, observational study. The proning practices adopted by our center may not be transferable to other intensive care units worldwide. The prone positioning procedure is resource-exhaustive and requires appropriately trained personnel to perform multiple prolonged cycles, which may not be feasible in resource-scarce settings. Although there were 931 prone episodes during the first two waves, we were only able to collect appropriate before and after prone oxygenation data for 776 prone episodes. This reflects the availability of real-world data in a pandemic setting. Nevertheless, we feel this cohort is representative of sicker COVID-19 patients who required prolonged mechanical ventilation with variations in timing and duration of prone positioning to improve their oxygenation at various times of their ICU journey. Another limitation is that, due to the study’s retrospective observational nature and inconsistent reporting, we were unable to present the rate of adverse events associated with prone positioning.

## 5. Conclusions

COVID-19 patients who require invasive mechanical ventilation have a high fatality rate. This observational study demonstrated a significant improvement in oxygenation after proning, even after 14 days from the initiation of mechanical ventilation. Proning cycles lasted 16 h, in accordance with current ARDS guidelines, and patients required a median of 5 cycles. Although the study shows that prone positioning improves oxygenation, larger, prospective randomized controlled trials are required to determine whether this improvement confers a survival advantage.

## Figures and Tables

**Figure 1 jcm-12-04226-f001:**
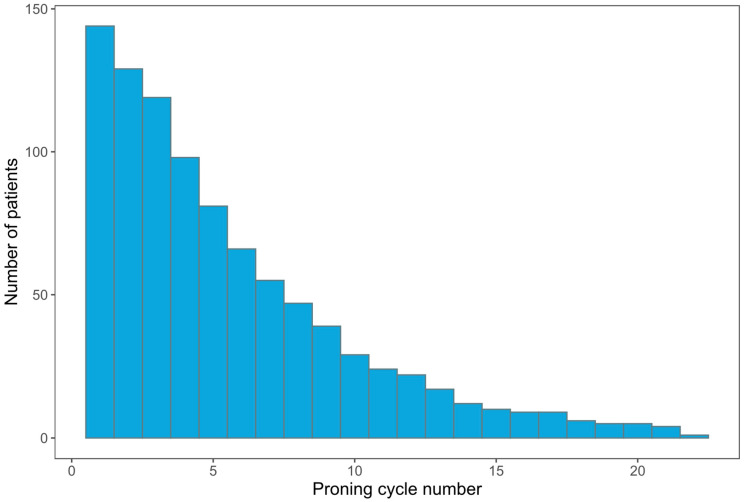
Bar chart view of the total number of prone cycles instituted for the number of patients included in the study.

**Figure 2 jcm-12-04226-f002:**
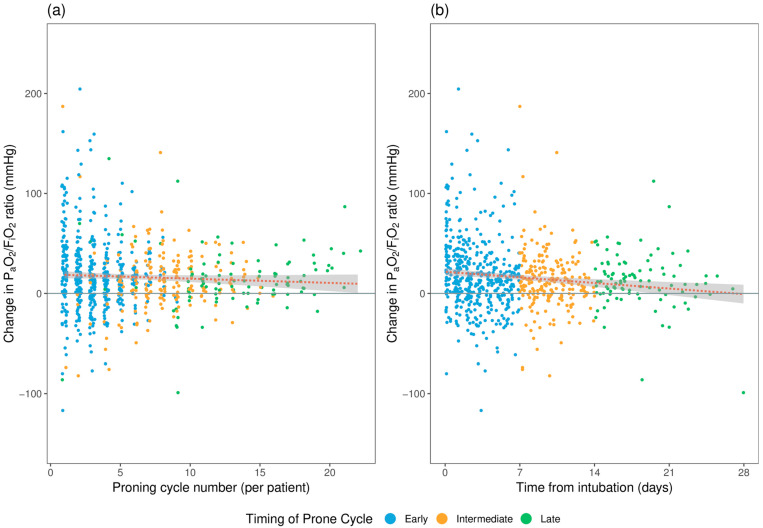
Demonstration of the change in the PaO_2_/FiO_2_ ratio (**a**) after each proning cycle per patient, (**b**) by prone cycle timing from time of intubation: early (<7 days), intermediate (7–14 days), and late (>14 days). The dotted line represents a linear model fitted to all plotted data. The shaded area represents the 95% CI.

**Table 1 jcm-12-04226-t001:** Demographic and baseline clinical data for mechanically ventilated patients with COVID-19 that use prone positioning as part of their clinical management.

Demographics	Invasive Mechanical Ventilation and Prone Position (n = 144)
Age, year	59 (51, 66)
Male sex, n (%)	95 (66%)
Duration of symptoms on admission to the ICU (days)	7.0 (6.0)
Admission APACHE II score	15 (11, 23)
Admission SOFA score	4 (3.0, 6.3)
Admission Charlson comorbidity index	2 (1, 3)
Admission PaO_2_/FiO_2_ (mmHg)	114.8 (87.0, 141.0)
Time to intubation from ICU admission (hours)	18.3 (1.6, 75.2)
BMI (kg/m^2^)	31 (26, 36)
Specific co-morbidities, n (%)	
BMI ≥ 30 (kg/m^2^)	79 (55%)
Diabetes Mellitus	45 (31%)
Chronic respiratory illness	29 (20%)
Ischemic heart disease	12 (8.3%)
Congestive cardiac failure	4 (2.8%)
Immunosuppression	14 (9.7%)
ICU admission blood	
Bilirubin (μmol/L)	10.0 (7.0, 14.0)
Creatinine (μmol/L)	73 (56, 97)
eGFR	88.5 (22.8)
Urea (mmol/L)	7.0 (5.5, 10.3)
CRP (mg/L)	138 (74, 209)
WBC (n × 10^9^/L)	8.7 (6.2, 11.4)
Lymphocytes (n × 10^9^/L)	0.70 (0.50, 0.90)
INR	1.10 (1.00, 1.25)
Ferritin (ng/mL)	832 (448, 1343)
HS troponin I (ng/L)	14 (8, 46)
Lactate dehydrogenase (U/L)	1031 (816, 1311)
D-Dimer (μg/L)	574 (299, 1121)
Creatine Kinase (U/L)	141 (66, 444)

Data presented by median (interquartile range) or number and percentages. APACHE II, Acute Physiology and Chronic Health Evaluation II score; BMI, body mass index; CRP, C-reactive protein; eGFR, estimated glomerular filtration rate; HS Troponin, High Sensitivity Troponin; INR, International Normalized Ratio; PaO_2_/FiO_2_, ratio of arterial oxygen partial pressure to fractional inspired oxygen; SOFA, Sequential Organ Failure Assessment score; WBC, white blood cell count.

**Table 2 jcm-12-04226-t002:** Proning cycle characteristics.

Proning Cycle Characteristics	All Proning Cycles	Early (<7 days)	Intermediate (7–14 days)	Late (>14 days)	*p*-Value
Total number of cycles, n	931	563	235	133	N/A
Number of patients proned	144	137	72	39	N/A
Cycles per patient *	5 (3, 9)	4 (3, 5)	3 (2, 4)	3 (1, 5)	NA
Duration of each cycle, hours *	16.0 (15.0, 17.0)	16.0 (15.0, 17.0)	16.0 (14.0, 17.0)	16.0 (14.0, 17.0)	0.007

Data are sub-grouped according to the timing of proning initiation, within 7 days of initiation of mechanical ventilation (early), 7–14 days (intermediate), or >14 days (late) from intubation. * Data are presented as median (IQR). Data are analyzed by Kruskal-Wallis’s rank sum test.

**Table 3 jcm-12-04226-t003:** The change in PaO_2_/FiO_2_ and PaCO_2_ before and after proning cycles for early (<7 days), intermediate (7–14 days), and late (>14 days).

Outcomes	Total Proning Cycles n = 776	Early (<7 days) n = 469	Intermediate (7–14 days) n = 203	Late (>14 days) n = 104	*p*-Value
Pre-prone FiO_2_ *	0.69 ± 0.16	0.68 ± 0.16	0.70 ± 0.15	0.71 ± 0.16	0.12
∆PaO_2_/FiO_2_ (mmHg) *	16.6 ± 34.4	18.5 ± 36.7	13.2 ± 30.4	14.8 ± 30.5	0.2
∆PaCO_2_ (mmHg) *	2.2 ± 7.4	2.0 ± 6.5	2.2 ± 8.8	2.9 ± 7.9	0.5

* Data are presented as mean ± standard deviation.

**Table 4 jcm-12-04226-t004:** Subgroup analysis for the pre-prone FiO_2_, ∆PaO_2_/FiO_2_ ratio, and ∆PaCO_2_ for each proning cycle.

	Frequency of Cycles ^†^	Pre-Prone FiO_2_ *	∆PaO_2_/FiO_2_ (mmHg) *	∆PaCO_2_ (mmHg) *
Cycle 1	108	0.72 ± 0.16	26.8 ± 48.4	2.8 ± 6.9
Cycle 2	113	0.63 ± 0.15	19.6 ± 39.6	1.2 ± 5.7
Cycle 3	100	0.61 ± 0.15	15.0 ± 39.7	3.0 ± 7.4
Cycle 4	81	0.70 ± 0.19	9.0 ± 32.4	2.9 ± 7.9
Cycle 5	67	0.69 ± 0.17	14.7 ± 26.7	1.6 ± 8.4
Cycle 6	52	0.74 ± 0.16	14.4 ± 25.6	2.1 ± 8.8
Cycle 7	45	0.72 ± 0.15	17.1 ± 24.5	1.5 ± 5.4
Cycle 8	38	0.70 ± 0.16	22.5 ± 29.6	0.6 ± 7.9
Cycle 9	35	0.71 ± 0.16	5.3 ± 33.2	4.8 ± 10.6
Cycle 10	25	0.70 ± 0.14	15.0 ± 17.9	1.9 ± 8.1

* Data are presented as mean ± standard deviation. ^†^ Total number of proning cycles where sufficient data were available to calculate the delta change.

**Table 5 jcm-12-04226-t005:** Clinical outcome of all mechanically ventilated proned patients (n = 144).

Outcome Characteristics	Patient Mechanically Ventilated and Proned n = 144
Duration of mechanical ventilation (hours)	397 (242, 746) *
30-day survival	104 (72%) **
Survival to hospital discharge	93 (65%) **

* Presented as median and interquartile ranges. ** Number of patients (%).

## Data Availability

All data are available at Jackson, Alexander; Dushianthan, Ahilanandan (2023): Prone positioning in mechanically ventilated COVID-19 patients. figshare. Dataset. https://doi.org/10.6084/m9.figshare.22643317.v1. (accessed on 17 April 2023).

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
