# Peer review of "Prone Positioning in Mechanically Ventilated COVID-19 Patients: Timing of Initiation and Outcomes"

_jcm, 2023, doi:10.3390/jcm12134226_

Round 1

Reviewer 1 Report

This manuscript describes a retrospective observational single-center study of subjects who were endotracheally intubated and mechanically ventilated for COVID-19 acute respiratory failure.  All received proning as part of their therapy.  Proning was associated with short-term increases in oxygenation and small increases in PaCO2.  No differences in oxygenation were seen among pronation sessions performed at various times during the subjects’ illness.

GENERAL COMMENTS:

1. The significance of the manuscript is limited by its study design (single-center retrospective observational study).  The study groups varied by the timing of initiation of proning therapy (early, intermediate, and length).  The main outcome variable was identified as a change in PaO2/FiO2 ratio before and after pronation.

2. The study is a part of a larger data collection effort by this group, which is described in reference 18.

3. What were the exclusion criteria for proning patients?  What were specific reasons for not proning patients?  Were patients proned that were not included in the data collection (e.g., missing or inadequate data, etc.)

4. What was the role of ECMO in your ICU protocols?  Was ECMO utilized in any patient?  Why or why not?  In discussion, how might your findings be useful in determining when a patient might benefit from the addition of ECMO therapy?

5. While carefully written, the manuscript appears to have been edited by multiple authors, which have introduced multiple small errors in grammar and sentence structure.  Some of these are listed as specific comments.  This reviewer does not intend to be “picky” by listing these changes, but only helpful for revision.  Please re-read you final effort carefully for such small errors that invariably creep into the editing process.

SPECIFIC COMMENTS:

Line 94: please provide a brief additional description or reference of your quality assurance and outlier management methods.

Line 123:  Add unit of years to age.

Lines 151-152: I could not find the abbreviation IMV in the table.  Please delete the description in the legend.

Lines 163-164: Check grammar.  I would recommend revising as: “Again, no difference was seen based on timing of proning cycle.”

Line 165: Figure 2 “demonstrates the change”

Line 169: “per patient. b) by prone cycle”

Lines 178-179: “with the most prominent improvement in oxygenation noted for the first proning episode”

Lines 201 and 202: “improvements” and “increases”

Line 209: “needed more proning sessions”

Line 231: “or above” rather than “or below.”  Correct?

Lines 253-254: “following rapid, extensive training”

Line 262: “settings”

Line 266: “sicker COVID-19 patients who”

Lines 269-270: I could not find where the criteria for collection of adverse events was described in the methods or where the incidence of inadvertent extubation was documented in the results section.  Please add these to the body of the manuscript.

Tables 1 and 5.  Center-justified columns are hard for the reader to follow.  Please left-justify column 1 and right-justify column 2.

While carefully written, the manuscript appears to have been edited by multiple authors, which have introduced multiple small errors in grammar and sentence structure.  Some of these are listed as specific comments.  This reviewer does not intend to be “picky” by listing these changes, but only helpful for revision.  Please re-read you final effort carefully for such small errors that invariably creep into the editing process.

Author Response

We are grateful and thank you for the reviewer's comments. Please see the following response to the reviewer's comments. 

  1. The significance of the manuscript is limited by its study design (single-center retrospective observational study).The study groups varied by the timing of initiation of proning therapy (early, intermediate, and length).  The main outcome variable was identified as a change in PaO2/FiO2 ratio before and after pronation.

Response: This was a pragmatic observational study and we have clarifed the limitations in the discussion section.  

  1. The study is a part of a larger data collection effort by this group, which is described in reference 18.

Response: Thank you. 

  1. What were the exclusion criteria for proning patients?What were specific reasons for not proning patients?  Were patients proned that were not included in the data collection (e.g., missing or inadequate data, etc.)

Response: No exclusion criteria applied as all of these patients were covid-19 and there were no patients with trauma, facial injuries, etc. The only limitation was the availability of prone team and we have clarified this in the method section now in lines 114-116. Although there were 931 prone sessions, we were only able to analyse 776 (line 193-194) due to missing data.

  1. What was the role of ECMO in your ICU protocols?Was ECMO utilized in any patient?  Why or why not?  In discussion, how might your findings be useful in determining when a patient might benefit from the addition of ECMO therapy?

Response: In the UK, ECMO is only provided by the specialist ECMO centres. We have added a paragraph in the methods section to clarify this further (106-108) 

  1. While carefully written, the manuscript appears to have been edited by multiple authors, which have introduced multiple small errors in grammar and sentence structure.Some of these are listed as specific comments.  This reviewer does not intend to be “picky” by listing these changes, but only helpful for revision.  Please re-read you final effort carefully for such small errors that invariably creep into the editing process.

Response: Thank you and we have modified the manuscript accordingly. 

SPECIFIC COMMENTS:

Line 94: please provide a brief additional description or reference of your quality assurance and outlier management methods.

Response: We have introduced a paragraph in the methods section to address this (131-145).

Line 123:  Add unit of years to age.

Response: Thank you, corrected as suggested. 

Lines 151-152: I could not find the abbreviation IMV in the table.  Please delete the description in the legend.

Response: Thank you, corrected as suggested. 

Lines 163-164: Check grammar.  I would recommend revising as: “Again, no difference was seen based on timing of proning cycle.”

Response: Thank you, corrected as suggested. 

Line 165: Figure 2 “demonstrates the change”

Response: Thank you, corrected as suggested. 

Line 169: “per patient. b) by prone cycle”

Response: Thank you, corrected as suggested. 

Lines 178-179: “with the most prominent improvement in oxygenation noted for the first proning episode”

Response: Thank you, corrected as suggested. 

Lines 201 and 202: “improvements” and “increases”

Response: Thank you, corrected as suggested. 

Line 209: “needed more proning sessions”

Response: Thank you, corrected as suggested. 

Line 231: “or above” rather than “or below.”  Correct?

Response: Thank you, this was a typo and we have corrected it now.

Lines 253-254: “following rapid, extensive training”

Response: Thank you, corrected as suggested. 

Line 262: “settings”

Response: Thank you, corrected as suggested. 

Line 266: “sicker COVID-19 patients who”

Response: Thank you, corrected as suggested. 

Lines 269-270: I could not find where the criteria for collection of adverse events was described in the methods or where the incidence of inadvertent extubation was documented in the results section.  Please add these to the body of the manuscript.

Response: As this was an obseravtional cohort, the adverse events reporting was not consistent. We have modified this sentence now. Thank you. 

Tables 1 and 5.  Center-justified columns are hard for the reader to follow.  Please left-justify column 1 and right-justify column 2.

Response: Thank you, corrected as suggested. 

Reviewer 2 Report

Dear Authors,

it's a great pleasure for me to review this well-written paper about the impact of initiation of prone positioning in COVID-19 patients receiving IMV.

The abstract and conclusions reflect the aims declared in the hypothesis.

Results are well written, I have only one question: did you try to stratify the patients according to age, underlying diseases or severity of hypoxemia before prone positiong (i.e., based on P/F ratio before pronation).

 Same concern about mortality.

I didn't understand if all patients received the same or similar ventilatory protocols (such as 'protective ventilation', high vs low PEEP ect ect).

Additionally, I recommend to add these interesting references on PEEP setting during supine and prone positioning in COVID-19 patients: PMID: 35876133, PMID: 35708999.

Thanks a lot.

Author Response

We are grateful for the reviewer's comments.

it's a great pleasure for me to review this well-written paper about the impact of initiation of prone positioning in COVID-19 patients receiving IMV.

Response: Thank you.

The abstract and conclusions reflect the aims declared in the hypothesis.

Response: Thank you. 

Results are well written, I have only one question: did you try to stratify the patients according to age, underlying diseases or severity of hypoxemia before prone positiong (i.e., based on P/F ratio before pronation).

Response: No, we did not perform multiple stratification for common variables as it will dilute the strength of the results and the robustness of the large oxygenation data set. 

Same concern about mortality.

Response: Thank you, the primary aim of this study is to evaluate the oxygenation response for individual prone episodes started at various timelines. 

I didn't understand if all patients received the same or similar ventilatory protocols (such as 'protective ventilation', high vs low PEEP ect ect).

Response: Thank you, We have introduced a new paragraph in the methods section to clarify this further (96-108).

Additionally, I recommend to add these interesting references on PEEP setting during supine and prone positioning in COVID-19 patients: PMID: 35876133, PMID: 35708999

Response: We have introdced a paragraph in the discussion section to include the refrences mentioned above (338-351.